# A versatilely high fidelity electric machines emulator for rapid testing of motor controller

**Yifeng Guo**[1], **Limin Huang**[1]*, **Min Zhang**[2], **Min He**[2], **Bin Zhong**[1], **Dakun Fan**[1]

**1** School of Mechanical Engineering, Chengdu University, Chengdu, China, **2** Hezhou University, Hezhou, China

* hlmsn155@163.com

**Data Availability Statement:** All relevant data are within the paper and its Supporting Information files.

**Funding:** Limin Huang received Funded:Sichuan Province Science and Technology Support

## Abstract

Electric machines emulators (EMEs) based on hardware-in-the-loop (HIL), which effectively act as emulators to mimic the actual motor behavior of Interior Permanent Magnet (IPM) machines. EME is frequently used to evaluate motor controller and motor control methodologies prior to development. The inverse magnetization motor model, which is used as the basis for real-time simulation in this paper's proposal for an electric machine emulator system based on HIL, uses FEA to create the motor model data. The nonlinear features of the motor may be successfully replicated with this motor model, and the accuracy of the electric machine emulator can be enhanced by using a straightforward and trustworthy motor controller. The real-time simulation tool typhoon HIL is used in the study to develop a hardware-in-the-loop simulation platform for an IPM electric machines emulator.

## 1 Introduction

IPM are widely used in new energy vehicles, aerospace, ship electric propulsion, wind power generation [1–3], because of their simple structure, low loss and convenient control [4,5]. Conventional motor testing requires the construction of mechanical test benches, and different parameter types of motors require the construction of special mechanical test pins according to the testing requirements, which require the use of high-power power supplies and consume a lot of electrical energy, and the testing process is tedious and labor-intensive [6,7].

The characteristics of the voltage and current at the motor's stator are the primary objective of a genuine motor test an electric IPM.

Machines emulator is composed of electronic components and a real-time simulation device running the motor model, which can modify the motor parameters and load according to the test requirements. The EME replaces the real motor connected to the motor controller under test, simulates the output characteristics of the motor under different operating conditions at the port, and completes the motor controller test safely and undamaged. In terms of the input port of the test devices, the EME resembles to a genuine motor. In addition, motor control scheme testing at the pre-development stage can help to decrease development time and faults, making electric machines emulator research highly significant for real-world industrial applications [8].

Program (2023YFQ0092), which provided support in terms of experimental setup. Chengdu Science and Technology Program (2022-YF05-01393-SN), which provided support in data collection and theoretical analysis. Sichuan College Students Innovation and Entrepreneurship Training Program (S202211079055), which Provided support in cultivating students' innovative abilities.

**Competing interests:** Enter: The authors have declared that no competing interests exist.

The phrase "virtual motor" was first used in 1998 by H.J. Slater to evaluate motor drives [9]. Slater did so in the same year. By comparing the simulation results of the motor simulation with the real motor working conditions, H. J.'s team finalized the design of the hardware structure of the EME and created the mathematical model of the motor in the d-q axis [10]. A built-in permanent magnet motor was employed as the simulation object in order to increase the motor's weak magnetic capacity and reduce the need for permanent magnets.

Based on the literature [11], the mathematical model of PMSM based on the d-q coordinate system is applied as the real-time simulation model to simplify the motor model and facilitate the real-time simulation of the electric machines mimicry. Yet since the model is an ideal motor model, it cannot precisely simulate the motor's actual output characteristics. It was determined to use the turning velocity, rotor flux references, and stator current as state variables from the dynamic mathematical model of the motor that was supplied in the literature [12] in the coordinate system. The model is separated by the Euler methodology, which makes the virtual motor model easy to estimate and computationally less demanding. However, the Euler procedure has challenges preserving the stability of the virtual motor simulation output since it stays away from magnetic saturation and parameter variations. The literature [13] investigates the limitations of the closed-loop current control-based electric machine emulator and proposes a linear regulator for the motor model and PI controller. In order to make the electric machine emulator's frequency domain characteristics more similar to those of the simulated target PMSM, the controller avoids closed-loop current control inside of the electric machine emulator; however, the electric machines emulator based on the linear regulator has a higher overshoot. There are some delays that are difficult to completely get away from. Coupled circuit-finite element co-simulation is used in the literature [14,15] to improve the fidelity of the PMSM model. This approach includes a joint parametric and external circuit simulation compared to the model with lumped parameters through real-time simulation of the finite element motor model, where the finite element equations and the circuit equations are synthesized into a matrix equation for a unified solution. This motor modeling process, however, is based on a finite element online simulation with huge data analyzing, poor real-time performance, low computing power, and difficulty applying current simulation gear. By using finite element analysis software to create a look-up table model based on the accurate motor qualities, the literature [16–18] supplies a thorough machine model. The motor simulator can successfully emulate the operation of the IPM motor thanks to the look-up table model, which corresponds to the motor in connection with the rotor position, current advance angle, and current and torque deviations. However, the modeling method requires the calculation of the inverse of the voltage input Flux linkage, which leads to unstable values, especially in drive simulation systems with switching devices. The literature [19] describes a field-programmable gate array (FPGA)-based high-bandwidth (>20 kHz) motor emulator (ME) prototype for an AC motor that can accurately simulate motor currents. The literature [20] uses a back-to-back converter design to connect the device under test (DUT) through the link impedance to achieve the development of a motor load simulation platform. The literature [21] proposed a common DC bus configuration ME system with only AC-DC regeneration simulator stage.

To solve the drawback of time-consuming finite element real-time simulation, analyzing the variation of d-q axis Flux linkage with d-q axis current as well as rotor position by finite element and applying the simulation results to the motor simulator by curve fitting function is an effective solution [22].

In order to solve the above problems, a nonlinear motor model with high fidelity, considering IPM magnetic saturation, space harmonics and cross-coupling is proposed in this paper, based on the Flux Linkages of the motor. The core parameters of the motor model are obtained from finite element calculations, and the nonlinear factors such as magnetic saturation, cross-

saturation, cross-coupling, and cogging torque are reflected by the nonlinear relationships of inductance, Flux Linkages, and torque with current and rotor position. Compared with other modeling approaches, the motor model proposed in this paper does not require FEA online simulation, and the nonlinear parameters of the motor are stored in tables that can be used off-line; and combines the advantages of simple mathematical model form to simplify the derivation of equations and reduce the amount of table data generated by FEA results, which is simple to implement and improves the model accuracy and simulation efficiency. In addition, a simple and reliable current tracking algorithm is used to control the inverter to accurately track the commanded current to ensure the speed and accuracy of the electric machine emulator current tracking. The real-time simulation device running the motor model is connected to the device under test, making up a HIL real-time simulation platform with simple and easy to implement features.

The organization of this paper is as follows. Section 2 presents an overview of the electric machines emulator system architecture. Section 3 presents the theoretical analysis and modeling process based on the Flux Linkages nonlinear motor model. Section 4 presents the electric machines emulator controller design, containing improved internal mode control and the Luenberger torque observer. Section 5 presents the real-time simulation of the proposed system as a means of validation, and the section contains the transient and steady-state experimental results are analyzed and discussed. Section 6 finally concludes this paper.

## 2 Overview of the electric machines emulator system

The nonlinear simulation model of a motor serves as the foundation for the electric machine emulator constructed during this research, which produces the needed current by sampling the port voltage as an input variable. This study refers to the EME of this mode as the VTC (Voltage to Current) mode as the three-phase inverter utilized to replicate the electrical characteristics of the motor is similar to a regulated current source in this case. The indicated electric machines emulator in this study aims to evaluate the motor controller and drive inverter in all respects in a lossless environment.

With the goal to emulate the port characteristics of the permanent magnet synchronous motor, Fig 1 shows a three-phase voltage inverter with base L filtering coupled to the driver that will be presented to the test. The rotor winding of the virtual motor is symbolized by the inverter circuit in the structural block diagram of the electric machines emulator, the induced electric potential is constituted by the capacitance of the filtered coupling circuit, the stator inductance is indicated by the inductance L of the filtered coupling circuit, and the internal resistance R of the inductor L is referred to by the virtual permanent magnet synchronous motor.

An architecture that mimics voltage input-current output is chosen for the electric machines simulation system that is suggested in this study. The motor model contains information obtained through finite element simulation and run in the form of an offline look-up table model that is related to the magnetic saturation of the machine, geometrical characteristics, spatial harmonics, etc. The motor modeling principles will be further discussed in Section III. Results of the motor simulation have a high degree of realism and precision since the process of simulation specified in the article employs a dynamic model of the motor. Additionally, a simple L-filter links the controller under test with the motor simulation device. Accurate command current acquisition and port control form the basis of the motor simulator in VTC mode. The proposed motor simulator system in the present study is set using a simple and durable current tracking control method. The next sections supply an explanation of the design process and analysis for the controller in an active control mode.

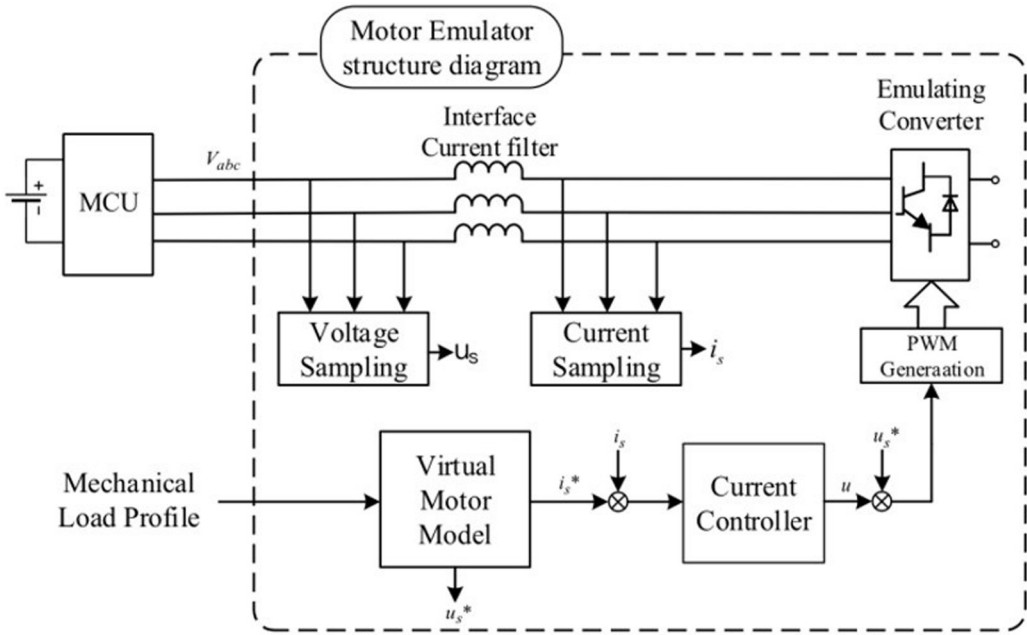

**Fig 1. Schematic of the proposed electric machine emulator system.**

## 3 Machine model for the proposed electric machine emulator

### Conventional PMSM model

The voltage and torque equations for the d-q axis, as described in Eq (1) and Fig 2, are a basis of the conventional modeling approach used for PMSM in the literature. A more standard magnetic circuit model, the conventional motor d-q axis model advantages include easy the

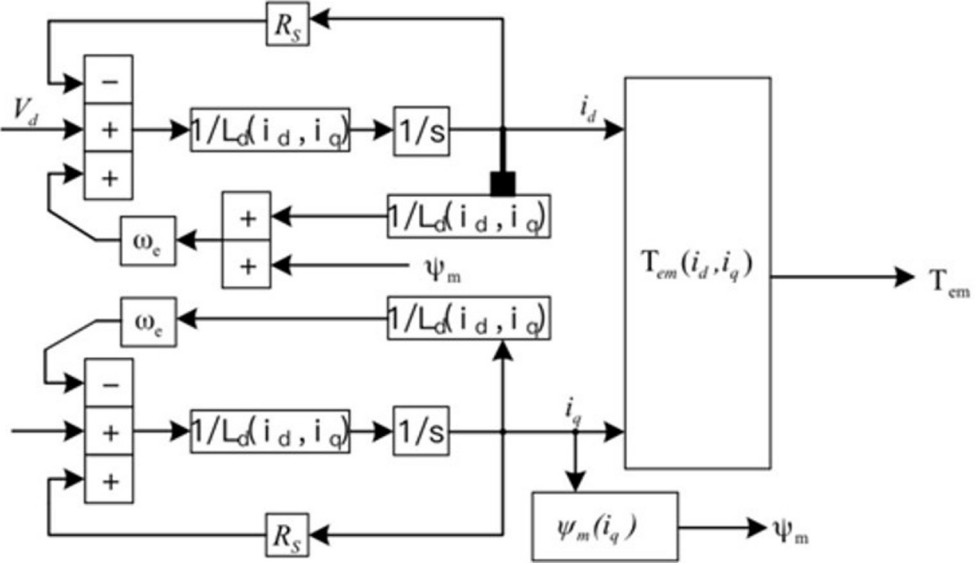

**Fig 2. Block diagram for modeling a conventional d-q coordinate system motor.**

calculation and a distinct relationship between motor the features.

$$u_d = R_{id} + \frac{d}{dx}\psi_d - \omega_e\psi_q$$

$$u_q = R_{iq} + \frac{d}{dx}\psi_q - \omega_e\psi_d$$

$$T_e = \frac{3}{2}p_n i_q \left[ i_d \left( L_d - L_q + \psi_f \right) \right] \tag{1}$$

$$J\frac{d\omega}{dt} = T_e - T_L - B\omega$$

Where $u_d, u_q, i_d, i_q$ are the voltages and currents in the d-q axis, $L_d, L_q$ are the self-inductances in the d-q axis. $\psi_f$ is the permanent Flux linkage, $R$ is the motor stator resistance, $\omega_e$ is the electric angular velocity, and $p$ is the number of pole pairs.

The cross-direct axis inductance parameters of the PMSM are subject to nonlinearity with the change in cross- and direct-axis currents as the conventional PMSM linear modeling is based on the voltage equation noted by self-inductance. The common linear motor model, on the contrary fingers, ignores the saturation of the magnetic circuit and the cross-coupling between parameters and only considers consideration of the air-gap Flux Linkages and the basic component of the inductance. As the result, the actual motor and such a motor model might not appear to be precisely the same. The permanent magnet synchronous motor is a high-order, multivariable, nonlinear, and complex system in actual operation, and the conventional linear model can no longer satisfy the requirements of a high-precision, high-performance motor simulator system. These effects consist of space harmonics, magnetic circuit saturation, cross-saturation, and cross-coupling. Further, the model and its state parameters are too flawless and the simulation time scale in off-line simulation technology does not precisely correspond the actual time, which makes it difficult bringing about the veracity and confidence of simulation conclusions.

## Conventional FEA PMSM model

$$\psi_a = \psi_{f,a} + \psi_{i,a} = \psi_{f,a} + \begin{bmatrix} L_{aa} & L_{ab} & L_{ac} \end{bmatrix} \begin{bmatrix} i_a \\ i_b \\ i_c \end{bmatrix}$$

$$v_a = Ri_a + \frac{d(L_{aa}i_a + L_{ab}i_b + L_{ac}i_c)}{dt} + \frac{d\psi_{f,a}}{dt} \tag{2}$$

$$\frac{dL_{aa}i_a}{dt} = L_{aa}\frac{di_a}{dt} + i_a \left( \frac{\partial L_{aa}}{\partial i_a}\frac{di_a}{dt} + \frac{\partial L_{aa}}{\partial i_b}\frac{di_b}{dt} + \frac{\partial L_{aa}}{\partial i_c}\frac{di_c}{dt} + \frac{\partial L_{aa}}{\partial \theta}\frac{d\theta}{dt} \right)$$

This common model is still mostly in differential form and is commonly referred to as the direct modeling a position of positive magnetization. With the linear motor model, the FEA model can obtain the Flux Linkages as long as the current and rotor position are included. This three-phase look-up table model utilizes a total of 36 differential expressions, three a variety of impedance tables, and nine offline tables. The model has an all-time high parameter the requirement and a poor model simulation efficiency.

## Proposed nonlinear PMSM Flux linkages model

In order to solve the above problems, a reverse magnetization PMSM model is proposed in this paper, based on magnetic flux as the state variable. The dynamic block diagram of the motor model is shown in Fig 3.

To describe it more intuitively, the equation of the Flux linkage is rewritten in matrix form. Meanwhile, to account for the cross-coupling effect between inductors, the effect of motor cross mutual inductance is added on this basis, and the matrix form of the Flux linkage equation is finally obtained as shown in Eq (3).

$$
\begin{bmatrix} \psi_d \\ \psi_q \end{bmatrix} = \begin{bmatrix} L_{dd} & L_{dq} \\ L_{qd} & L_{qq} \end{bmatrix} \begin{bmatrix} i_d \\ i_q \end{bmatrix} + \psi_f \cdot \begin{bmatrix} 1 \\ 0 \end{bmatrix}
\tag{3}
$$

Thus, the voltage equation can be written in the form of Eq (4).

$$
\begin{cases}
u_d = R_s i_d + L_{dd}\dfrac{di_d}{dt} + L_{dq}\dfrac{di_q}{dt} - \omega\left(L_{qq}i_q + L_{qd}i_d\right) \\[2mm]
u_q = R_s i_q + L_{qq}\dfrac{di_q}{dt} + L_{qd}\dfrac{di_d}{dt} - \omega\left(L_{dd}i_d + L_{dq}i_q + \psi_f\right)
\end{cases}
\tag{4}
$$

Where the inductance parameters—which are time-varying parameters—are obtained by solving the finite element method which involve the resistance, the permanent Flux linkage, the straight-axis synchronous inductance, the cross-axis synchronous inductance value, the mutual inductance of the cross-axis in the straight-axis, and the mutual inductance of the straight-axis in the cross-axis. The relationship between the inductance and the currents in the cross and straight axes is described through establishing a two-dimensional look-up table while on implementation.

To determine the parameters considering the spatial harmonics, the FEA results, including the rotation angle $\theta$ are applied to the PMSM model. Thus, the self-inductance model can be expressed as a function of $\theta$. Therefore, the voltage equation can be rewritten as in Eq (5).

$$
\begin{bmatrix} u_d \\ u_q \end{bmatrix} = R_s \begin{bmatrix} i_d \\ i_q \end{bmatrix} + \begin{bmatrix} \dfrac{d\varphi_d\left(i_d, i_q, \theta\right)}{dt} \\[3mm] \dfrac{d\varphi_q\left(i_d, i_q, \theta\right)}{dt} \end{bmatrix} + \omega_r \begin{bmatrix} -\varphi_q\left(i_d, i_q, \theta\right) \\[2mm] \varphi_d\left(i_d, i_q, \theta\right) \end{bmatrix}
\tag{5}
$$

Thus, the effects of both magnetic saturation and spatial harmonic fields are inherently

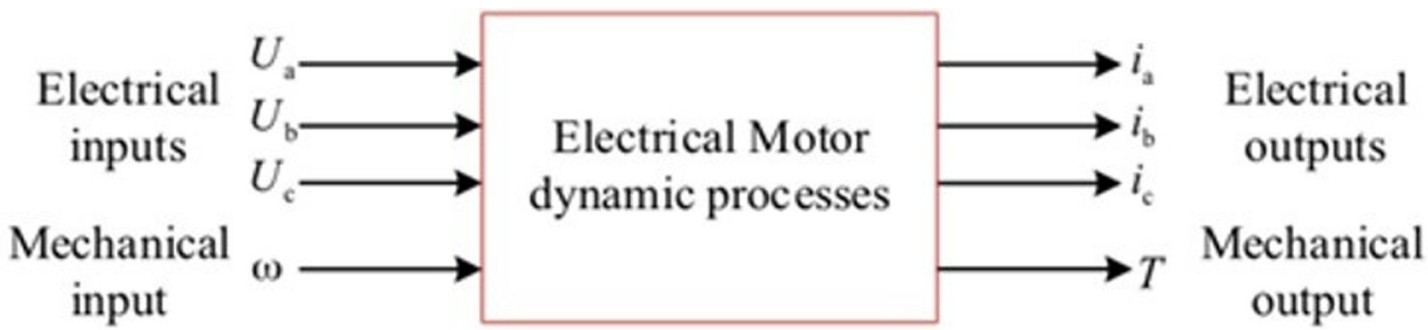

**Fig 3. Dynamic process of the motor.**

included in the above Flux linkage function. The dynamics equation of the motor can be simplified to Eq (6).

$$u = Ri + \omega\psi + \frac{d\psi}{dt} \tag{6}$$

Where $u, i$ and $\psi$ are the vectors of voltage and Flux linkage in the d-axis, q-axis and magnetic field. $R$ and $\omega$ are the matrices of resistance and electrical angular velocity.

The incremental inductance matrix $L$ consists of a self-inductance matrix $L_{self}$ and a mutual inductance matrix $L_{mutual}$, which as shown in Eq (7).

$$L_{self} = \begin{bmatrix} L_{dd} & 0 \\ 0 & L_{qq}^m \end{bmatrix}, L_{mutual} = \begin{bmatrix} 0 & L_{dq} \\ L_{qd} & \theta \end{bmatrix} \tag{7}$$

The incremental matrix $L$ is the sum of the self-inductance matrix and the mutual inductance matrix.

The voltage equation is rewritten as Eq (8).

$$\frac{d\psi}{df} = u - Ri - \omega\psi \tag{8}$$

The inverse of the Flux linkage can be further converted to the inverse of the current, which as shown in Eq (9).

$$\frac{d\psi}{dt} = L\frac{di}{dt} \tag{9}$$

The inverse of the inductance matrix is used to determine the inverse of the inductance, which as shown in Eq (10).

$$\frac{di}{dt} = L^{-1}\frac{d\psi}{dt} \tag{10}$$

The current can be obtained by integrating the current derivative, which as shown in Eq (11).

$$i = \int \frac{di}{dt}dt \tag{11}$$

Since the cross and straight axis mutual inductance is considered, the torque equation is shown in Eq (12).

$$T_e = \frac{P_{out}}{\omega_r} = \frac{3}{2}p\left(\psi_d i_q + \psi_q i_q\right) \tag{12}$$

The voltage equation of a conventional motor can be expressed in multiply differential or integral form, and its numerical calculation takes the form shown in Eq (13).

$$v = Ri(\psi, \theta) + \frac{d\psi}{dt}$$
$$\psi = \int (v - Ri(\psi, \theta))dt \tag{13}$$

Based on the above equation, this paper modeling in the integral form first needs to inverse the relationship to get, which is called the inverse magnetization modeling method, and its simulation is faster and more accurate compared to the differential form. Compared with the traditional linear model, the chain-based inverse magnetization PMSM model is shown in Fig 4 can reflect the nonlinear characteristics of the motor magnetic circuit saturation, cross-saturation,

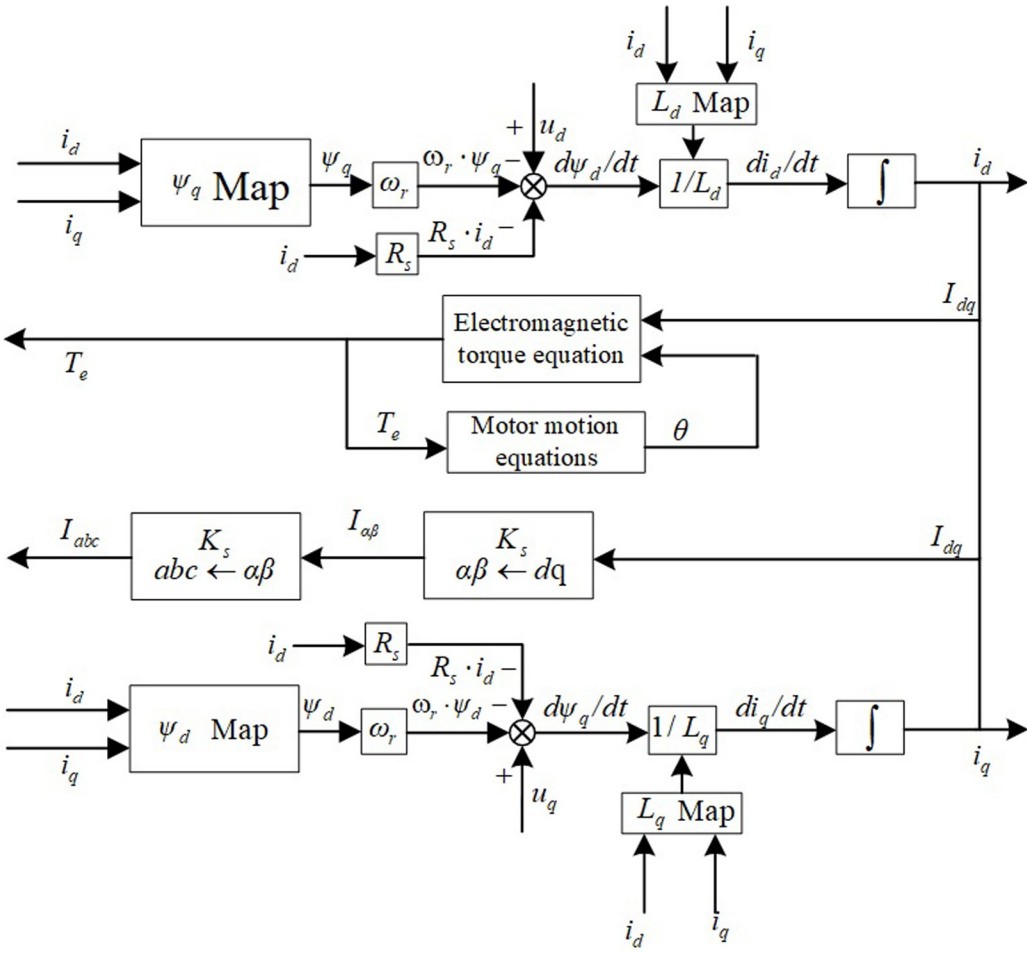

**Fig 4. Inverse magnetization PMSM model based on Flux linkage.**

cross-coupling, tooth slot torque, etc.; compared with the three-phase finite element look-up table model, the motor Flux linkage model only requires two Flux linkages and two inductance tables, which greatly reduces the number of parameters required for the model, and combines the advantages of simple mathematical model form with the real-time.

Also it fits real-time with advantages of basic mathematical models, improving the effectiveness of model simulation estimates. The motor model in the d-q coordinate system is easier to operate on than the three-phase model, making the HIL system easy and simple to implement.

## Nonlinear IPM Flux Linkages model finite element analysis

As shown in Fig 5, a FEA model of the three-phase 8-pole 48-slot IPM was built using the electromagnetic field FEA set max well to create the table look-up model. Table 1 displays the various motor training places, which data comes from Toyota's open source motor data. The 1/8 model is utilized for FEA in the present work, and the material definition, mesh division, and boundary conditions are specified. This is done bearing consideration of the symmetry of the motor model and the two-dimensional simulation length of time.

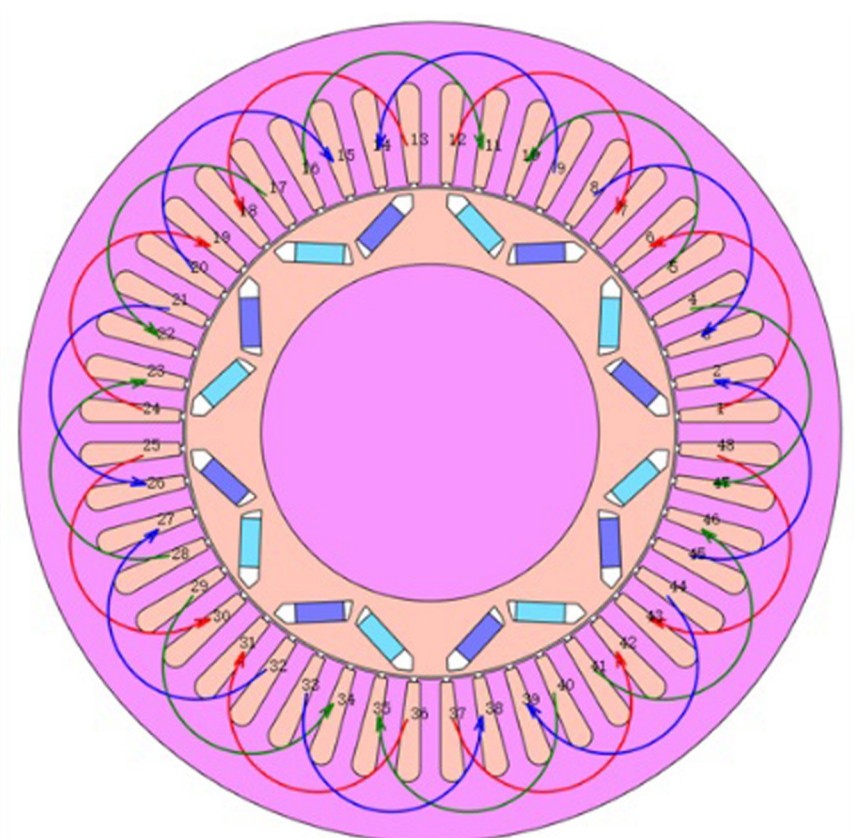

**Fig 5. PMSM finite element analysis model.**

The Flux linkages in the d-q axis are all obtained by a finite look-up table model with equations in Eqs (14) and (15).

$$\begin{cases} \psi_d = f\left(i_d, i_q, \theta\right) \\ \psi_q = g\left(i_d, i_q, \theta\right) \end{cases} \tag{14}$$

$$\begin{cases} L_d = H\left(i_d, i_q, \theta\right) \\ L_q = C\left(i_d, i_q, \theta\right) \end{cases} \tag{15}$$

Phase current amplitude, phase, and rotor position angle are defined as parametric analysis

**Table 1. IPM Motor parameters.**

| Parameter | Value | Parameter | Value |
|---|---|---|---|
| Nominal voltage | 400V | Maximum power | 60KW |
| Pole pair | 4 | Nominal speed | 2000rpm |
| Rotors outer diameter | 269.24mm | Stator outer diameter | 270 |
| Rotors inner diameter | 160.4mm | Stator inner diameter | 162 |
| WidthMag | 32mm | ThickMag | 6.49mm |

objects in the electromagnetic field solution set, at step sizes in 20 A, 15˚, and 3˚ and ranges of [-10A, 10A], [0,360˚], and [0,90˚], respectively. It enables the mapping relationship between flux linkage, torque, and phase current amplitude and phase to be calculated based on different rotor position angles. For the goal of discovering the three-dimensional table search model of flux linkage-current and torque-current under various rotor position angles, the solving range of FEA and is set to [-10A, -10A], and the solving step is 12 A.

The finite element calculation yields three finite element inductance matrix models of the built-in permanent magnet synchronous motor and as shown in Fig 6. For the rotor position angle $\theta$ = 30, the chain-current look-up table model is shown in Figs 7 and 8.

A three-dimensional look-up table model of the d-axis inductance versus the d-q axis current at the specific rotor position is displayed in Fig 7, and an analogous path of the q-axis inductance versus the d-q axis current at the same rotor position is shown in Fig 8. The seventh picture shows the magnetic saturation effect, while Fig 8 shows the spatial harmonics and flux chain torque. Torque ripples exist even in the case of constant d- and q-axis currents or sinusoidal phase currents; however, they cannot be predicted by conventional linear models, in spite of being present in real IPM machines.

A three-dimensional look-up table model of the d-axis flux linkage and the d-q axis current at a given rotor position is shown in Fig 9, and an identical model of the q-axis flux linkage and the d-q axis current at the same rotor position is shown in Fig 10. Current and flux linkage map to the other monotonically. The building of the current-flux inverse look-up table model has a mathematical basis due to the capability to determine a unique () for a given.

## 4 Proposed control design for the electric machines emulator

### Proposed control design for internal model controller

The motor port current is a non-repetitive transient component during the switching process when the motor operating state changes. The traditional PI controller cannot track the port current in real-time, and there is a steady-state error that is hard to eliminate, making it difficult to achieve the best control effect.

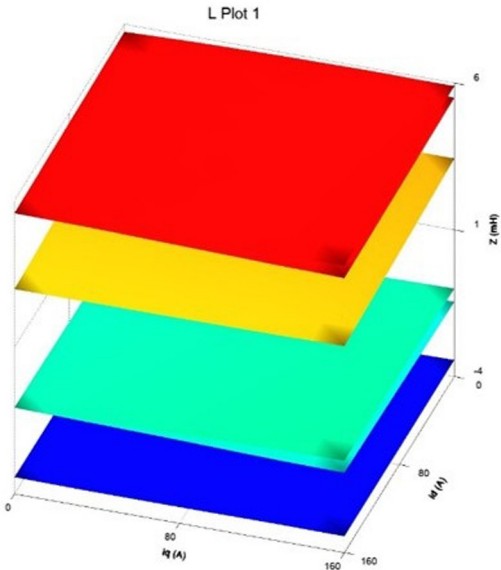

**Fig 6. Three-dimensional inductor look-up table model.**

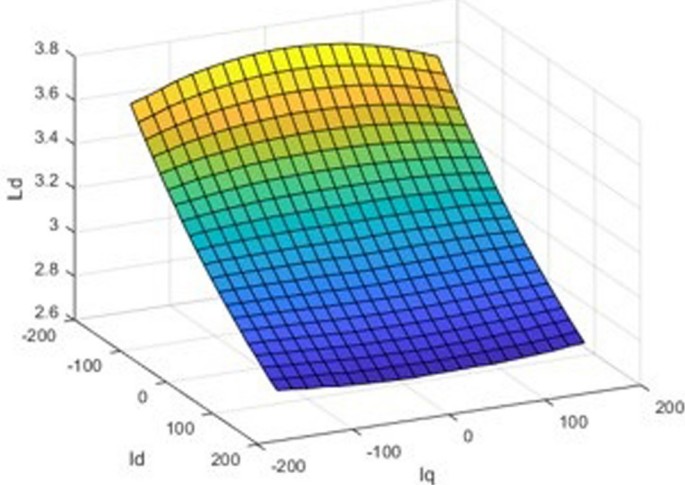

**Fig 7. d-axis inductance-current three-dimensional look-up table model.**

A model-based control strategy called internal mode control has been developed for chemical applications. Internal-mode control has a greater suppression of motor parameter determine errors because it comprises mathematical models that forecast the effects brought on by the control output and is insensitive to parameter swings. As a consequence, through integrating the enhanced internal-mode control into the permanent magnet synchronous motor control, the steady-state error created during current tracking can be minimised.

The internal model feedback controller structure is shown in Fig 11, where $R(s)$, $U(s)$, and $Y(s)$ correspond to the motor current, stator voltage, and stator current, respectively. $C(s)$ is the controller, $G(s)$ is the real model of the control object, and $\hat{G}(s)$ is the observed model. Assuming that the observed model is equivalent to the real model, i.e., $G(s) = \hat{G}(s)$, the output follows

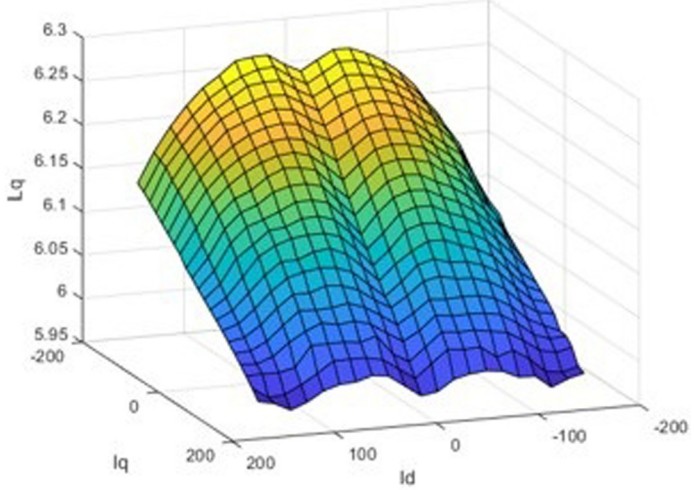

**Fig 8. q-axis inductance-current three-dimensional look-up table model.**

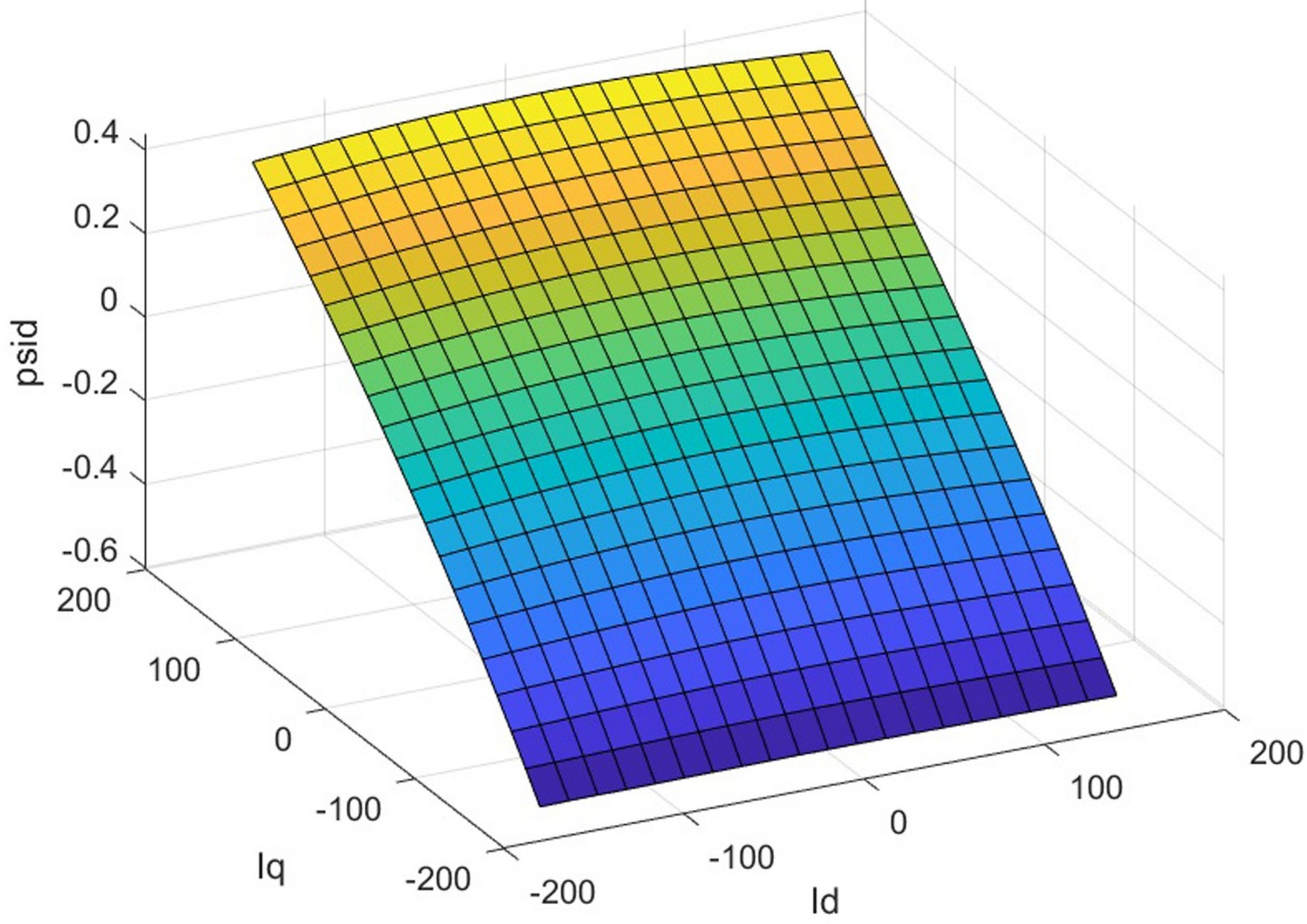

**Fig 9. d-axis Flux linkage-current three-dimensional look-up table model.**

the input exactly and the feedback signal has a value of 0.

$$F(s) = [I - C(s)\hat{G}(s)]^{-1}C(s) \tag{16}$$

Eq (16) only $C(s)$ is an unknown quantity. To make the system steady-state error 0, Eq (18) needs to be satisfied.

$$I - C(0)\hat{G}(0) = 0 \tag{17}$$

It can be taken as $C(s) = G^{-1}(s)$. The robustness of the system is improved by connecting the low-pass filter $L(s)$ in series, then $C(s)$ can be expressed by Eq (18).

$$C(s) = G^{-1}(s)L(s) \tag{18}$$

To simplify the design and improve the filtering performance, a low-pass filter with a first-

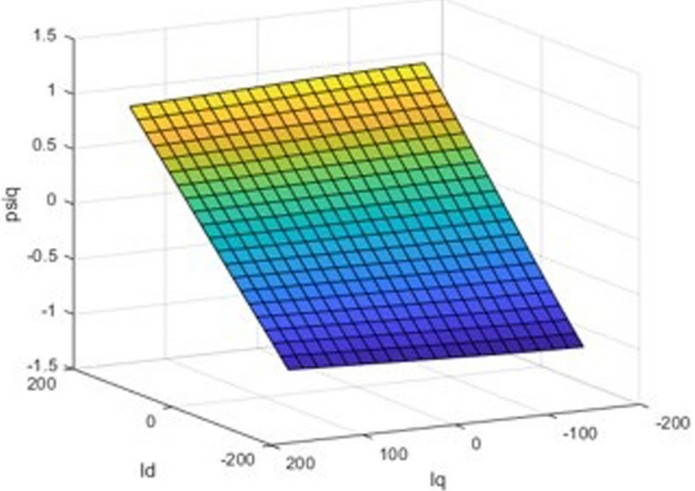

**Fig 10. q-axis Flux linkage-current three-dimensional look-up table model.**

order inertial link is selected, and which is shown Eq (19), with I as the unit matrix.

$$L(s) = \frac{\lambda}{\lambda + s} I \tag{19}$$

The adjusted internal mode controller can be expressed by Eq (20).

$$F(s) = [I - L(s)]^{-1} G^{-1}(s) L(s) \tag{20}$$

Substituting the d-q voltage state space equation into the voltage equation of its Laplace

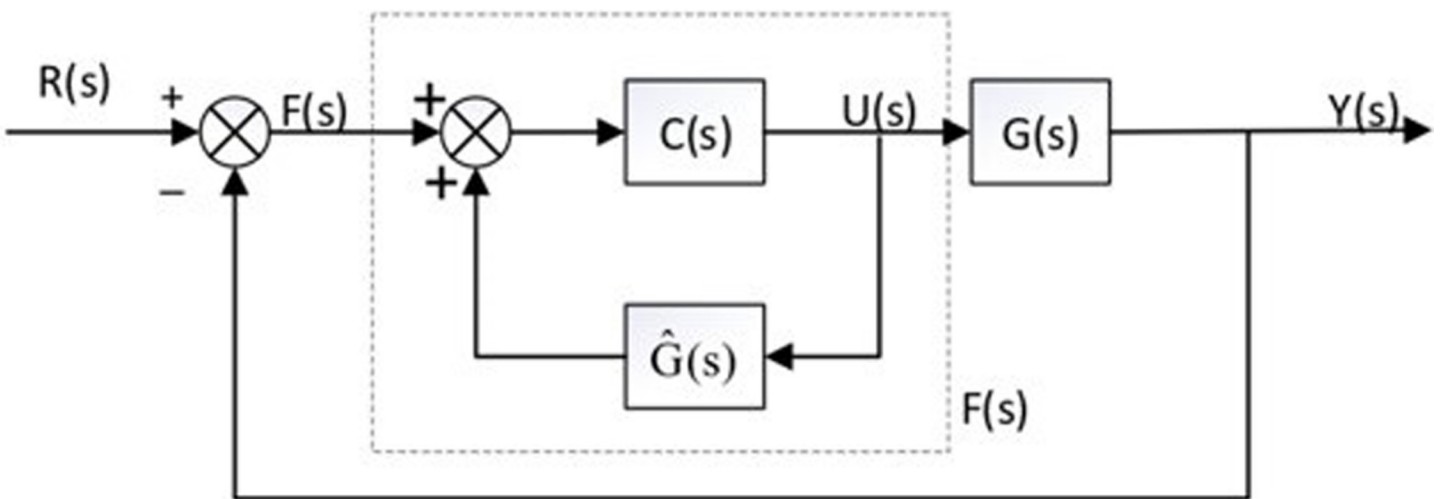

**Fig 11. Internal mode feedback controller.**

change, the internal model controller $F(s)$ can be obtained, as shown in Eq (21).

$$F(s) = \lambda \begin{bmatrix} R_s + sL_d & 0 \\ 0 & R_s + sL_q \end{bmatrix} \qquad (21)$$

$L_d$, $L_s$, and $R_s$ are the actual motor parameters, and $\lambda$ is an adjustable parameter.

Define $t_r$ as the time required for the system to complete the step, then the relationship between $t_r$ and $\lambda$ is shown in Eq (22).

$$t_r = \frac{ln9}{\lambda} \qquad (22)$$

The internal-mode control regulation parameter is only $\lambda$. The performance of both the current controller and the decoupling network is limited by the selection of $\lambda$. The initial design of the internal mode controller only considered the decoupling capability of the current, and its design did not consider the impact on the rapidity and accuracy of current tracking. Because of the linear relationship between the controller proportional and integral coefficients, and the regulation parameter is only $\lambda$, resulting in a small adjustable range of controller parameters, it is difficult to achieve the best control effect.

To solve the problem that the current controller and the decoupling network in the internal mode controller are difficult to reach the optimum at the same time, the internal mode controller is improved and the new internal mode controller is shown in Eq (23) and Fig 12.

$$F(s) = \begin{bmatrix} \lambda_1 R_s + \lambda_2 sL_d & 0 \\ 0 & \lambda_1 R_s + \lambda_2 sL_q \end{bmatrix} \qquad (23)$$

The parameter rectification time can be set first $\lambda_1 = \lambda_2 = \lambda$ using Eq (23) to meet the decoupling requirement of the internal-mode controller, and then the control performance and freedom of the controller can be improved by adjusting $\lambda_1$ and $\lambda_2$. The new internal-mode controller improves the freedom and flexibility of parameter adjustment, and there is no linear relationship between the proportional and integral coefficients in the controller, which improves the rapidity and stability of current tracking and makes the control and decoupling performance of the controller optimal.

## Proposed control design for the luenberger torque observer

Torque fluctuations in permanent magnet synchronous motors can result from vector control position detection errors, current detection errors, errors produced by the inverter, and motor features including Flux linkage harmonics and cogging effects. By sending the observed torque forward to the current loop to provide high frequency pulsing torque to counteract the variations in motor torque, the Lomborg torque observer may be constructed to smooth the system speed.

Since the system's inputs and outputs are discernible, the Lomborg observation utilizes these in order to recreate the observed values of the state variables.

The equation of mechanical motion of the PMSM is Eq (24).

$$J\frac{d\omega_m}{dt} = T_e - T_L - B\omega_m \qquad (24)$$

The load torque is not measurable and the speed is measurable. Selected: state quantity

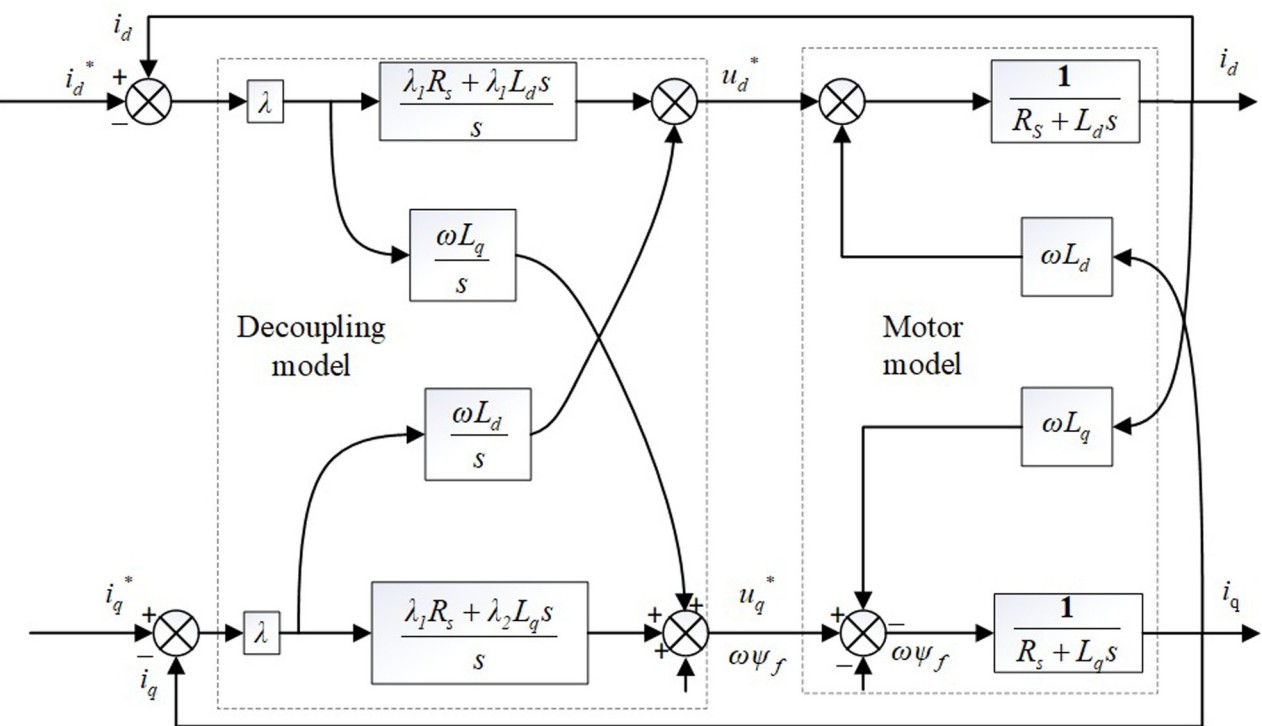

**Fig 12. Block diagram of improved internal mode control structure.**

$x = \begin{bmatrix} \omega_m \\ T_l \end{bmatrix}$, input quantity $x = \begin{bmatrix} \omega_m \\ T_l \end{bmatrix}$ and output quantity $y = \omega_m$. The general form of its state space is Eq (25).

$$\dot{x} = A^*x + B^*u, y = C^*x + D^*u \tag{25}$$

Eq (25) is replaced by a state space expression. When the control frequency is high and the sampling period is small, it is assumed that the torque is constant in each sampling period to obtain Eq (26).

$$\begin{bmatrix} \dot{\omega}_m \\ \dot{T}_l \end{bmatrix} = \begin{bmatrix} \dfrac{-B}{J} & \dfrac{-1}{J} \\ 0 & 0 \end{bmatrix} \begin{bmatrix} \omega_m \\ T_l \end{bmatrix} + \begin{bmatrix} \dfrac{1}{J} \\ 0 \end{bmatrix} T_e$$

$$y = \begin{bmatrix} 1 & 0 \end{bmatrix} \begin{bmatrix} \omega_m \\ T_l \end{bmatrix} \tag{26}$$

Where the A matrix is $\begin{bmatrix} \dfrac{-B}{J} & \dfrac{-1}{J} \\ 0 & 0 \end{bmatrix}$, the B matrix is $\begin{bmatrix} \dfrac{1}{J} \\ 0 \end{bmatrix}$, the C matrix is $\begin{bmatrix} 1 & 0 \end{bmatrix}$, and the D matrix is 0. The Romberg observer is as in Eq (27).

$$\dot{\hat{x}} = (A - LC)*\hat{x} + (B - LD)*u + L*y \tag{27}$$

From Eqs (26) and (27) is rewritten to obtain Eq (28).

$$\begin{bmatrix} \dot{\hat{\omega}}_m \\ \dot{\hat{T}}_l \end{bmatrix} = \begin{bmatrix} \dfrac{-B}{J} - L_1 & \dfrac{-1}{J} \\ -L_2 & 0 \end{bmatrix} \begin{bmatrix} \hat{\omega}_m \\ \hat{T}_l \end{bmatrix} + \begin{bmatrix} \dfrac{1}{J} \\ 0 \end{bmatrix} T_e + \begin{bmatrix} L_l \\ L_2 \end{bmatrix} \omega_m \tag{28}$$

Transform Eq (28) into Eq (29).

$$\dot{\hat{\omega}}_m = \frac{1}{J} * \left( T_e - \hat{T}_l - B^* \hat{\omega}_m + J * L_1 * (\omega_m - \hat{\omega}_m) \right)$$
$$\dot{\hat{T}}_l = L_2 * (\omega_m - \hat{\omega}_m) \tag{29}$$

In order to make the observer converge and stabilize as soon as possible and the deviation between the estimated value of the state variable and the actual value approaches zero, the solutions of the characteristic equation of the observation system must be negative. The characteristic equation of the observer is shown in Eq (30).

$$|\lambda E - (A - LC)| = \lambda^2 + \left( \frac{b}{J} + L_1 \right) \lambda + \frac{L_2}{J} = 0 \tag{30}$$

Assuming that the eigenvalues of the observer system are $\alpha$ and $\beta$, the feedback gain matrix (31) is obtained.

$$\lambda^2 - (\alpha + \beta)\lambda + \alpha\beta = 0$$
$$\begin{cases} L_1 = -(\alpha + \beta) - \dfrac{B}{J} \\ \\ L_2 = -\alpha\beta J \end{cases} \tag{31}$$

From Eq (29) we can then build the Romberg torque observer.

## 5 Experiment and analysis

### Experimental setup

In this paper, the real-time simulation device typhoon is used as the core of the real-time simulation and control of the motor simulator, and the real-time simulation of the output characteristics of the motor port is realized by the real-time simulation device typhoon. The experimental platform of the permanent magnet synchronous motor simulator consists of a PC host computer, DIM100 control board and real-time simulation device typhoon The structure of the permanent magnet synchronous motor simulation experimental platform is shown in Fig 13 [23,24].

Typhoon, a powerful multi-core processor machine platform, is coupled with real-time simulation technology. In order to perfectly serve the power electronics sector of simulation and testing, Typhoon features particular authority electronics model library, a professional modeling tool suite, high-precision real-time solution algorithms, and a high-speed I/O interface. Typhoon, a real-time simulation tool, characteristics an integrated FPGA solver that processes data in parallel with the system CPU, greatly improving the effectiveness of simulation operations.

It's significant to note that this is the first time that Typhoon HIL has been used to the motor simulator design research. This makes the design more complicated. Typhoon HILL is widely utilized in smart grids, highlighting that it is suitable for usage in new research fields.

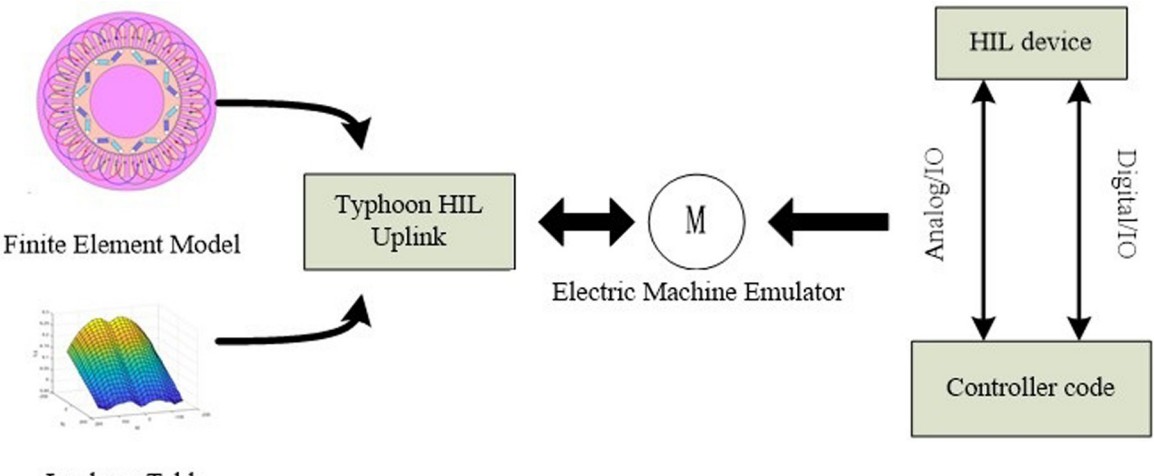

**Fig 13. Structure of the experimental platform of PMSM emulator.**

### Electric machines emulator steady state simulation analysis

The rated power is the motor output power under the best working condition, which is the highest efficiency of the motor, and the rated power is also the working condition recommended by the motor manufacturer, so the rated power can be understood as the most commonly used power of the motor. The purpose of this paper is to study a permanent magnet synchronous motor simulator with high accuracy and stability, all the parameters are chosen from the rated parameters of the motor, and only the transient and steady-state operating conditions of the motor are analyzed experimentally without considering the ultimate operating conditions. The parameters of the motor body are still selected from Table 1. The rated power is 60 KW and the rated speed is 2000 rpm, so the rated load torque is 30 $N \cdot m$. The motor simulator is used for steady-state experiments at 0 $N \cdot m$, 10 $N \cdot m$, 20 $N \cdot m$, and 30 $N \cdot m$ motor load torque, respectively.

Start the simulation in the host interface of the real-time simulation device typhoon, apply load torque after the motor starts, and set the motor speed to the rated speed of 2000 rpm. Observe the real-time waveform comparison of the AC/DC current and angular velocity of the permanent magnet synchronous motor simulator and the motor. The red line represents the real-time simulation waveform of the motor simulator, and the black line represents the real-time simulation waveform of the real motor module command value.

From Figs 14–17, it is found that the output of the PM synchronous motor simulator follows the commanded values of load and speed better. Comparing the real-time simulation waveform of the direct shaft current $i_d$ of the PM synchronous motor simulator and the motor module, the waveform of the motor simulator is slightly more unstable, but the waveform oscillation of the direct shaft current of the motor simulator is ±0.1 A. The direct shaft current of the motor simulator represented by the red waveform is basically the same as that of the actual motor module in terms of value, and the direct shaft current $i_d$ of the four loads of the motor simulator is stable at 0.3 A. The relative errors are -0.017%, 0.082%, 0.002%, and -0.001%, respectively, which are all less than 5% and meet the requirements of motor control strategy testing.

Observe the real-time simulation waveform of the cross-axis current $i_q$ in Figs 14–17. Since the motor is given a load, the cross-axis current is stable at about 0 A, 0.3 A, 0.625 A, and 0.925

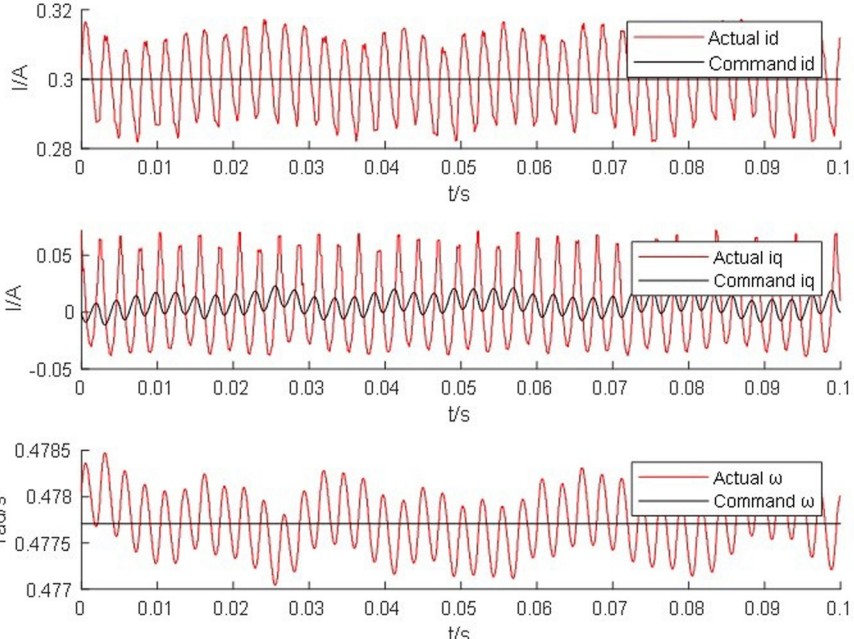

**Fig 14. Motor simulator 0$N\cdot m$ hardware-in-the-loop simulation waveform.**

A, respectively, and the waveform oscillation of the cross-axis current is ±0.05 A. As the load increases, the value of the cross-axis current $i_q$ increases. The values of the cross-axis current of the motor simulator are basically consistent with the motor module, and the relative errors are 4.940%, -0.012%, -0.012%, and 0.0723%, respectively, which are all less than 5% and meet the requirements of the motor control strategy test.

Observe the real-time simulation waveforms of angular velocity in Figs 14–17. Since the output of the actual motor can only output the rotor angular velocity, comparing the rotor angular velocity of the simulated motor with that of the actual motor, the angular velocity is stable at 0.4785 rad/s, 0.478 rad/s, 0.478 rad/s and 0.4778 rad/s under the four torque loads; the four The speed values shown on the oscilloscope are the standard values, and the speed of the motor simulator under four loads are 2005.995 rpm, 2007.091 rpm, 2003.891 rpm and 2003.060 rpm after conversion, which are basically consistent with the speed command value of 2000 rpm. The relative errors are 0.3%, 0.355%, 0.195% and 0.153% respectively, which are less than 5%, and the speed of the motor simulator meets the control effect requirements.

The Steady State Simulation results are shown in Table 2. From the analysis of the simulation waveform of steady-state operation, it can be concluded that under the same motor parameters, control strategy and circuit structure, the permanent magnet synchronous motor simulator can accurately simulate the port output characteristics of the real motor and simulate the condition of the real motor driving the load in steady-state operation.

## Electric machines emulator transient simulation analysis

The motor speed was maintained at the rated speed of 2000 rpm while the torque load was increased from 0 to 10, 10 to 20, and 20 to 30 during the steady-state operation of the motor simulator, respectively, to simulate and observe the transient operation of the motor during sudden load addition. This was done after providing the accuracy and stability of the steady-state operation of the permanent magnet synchronous motor simulator in the previous

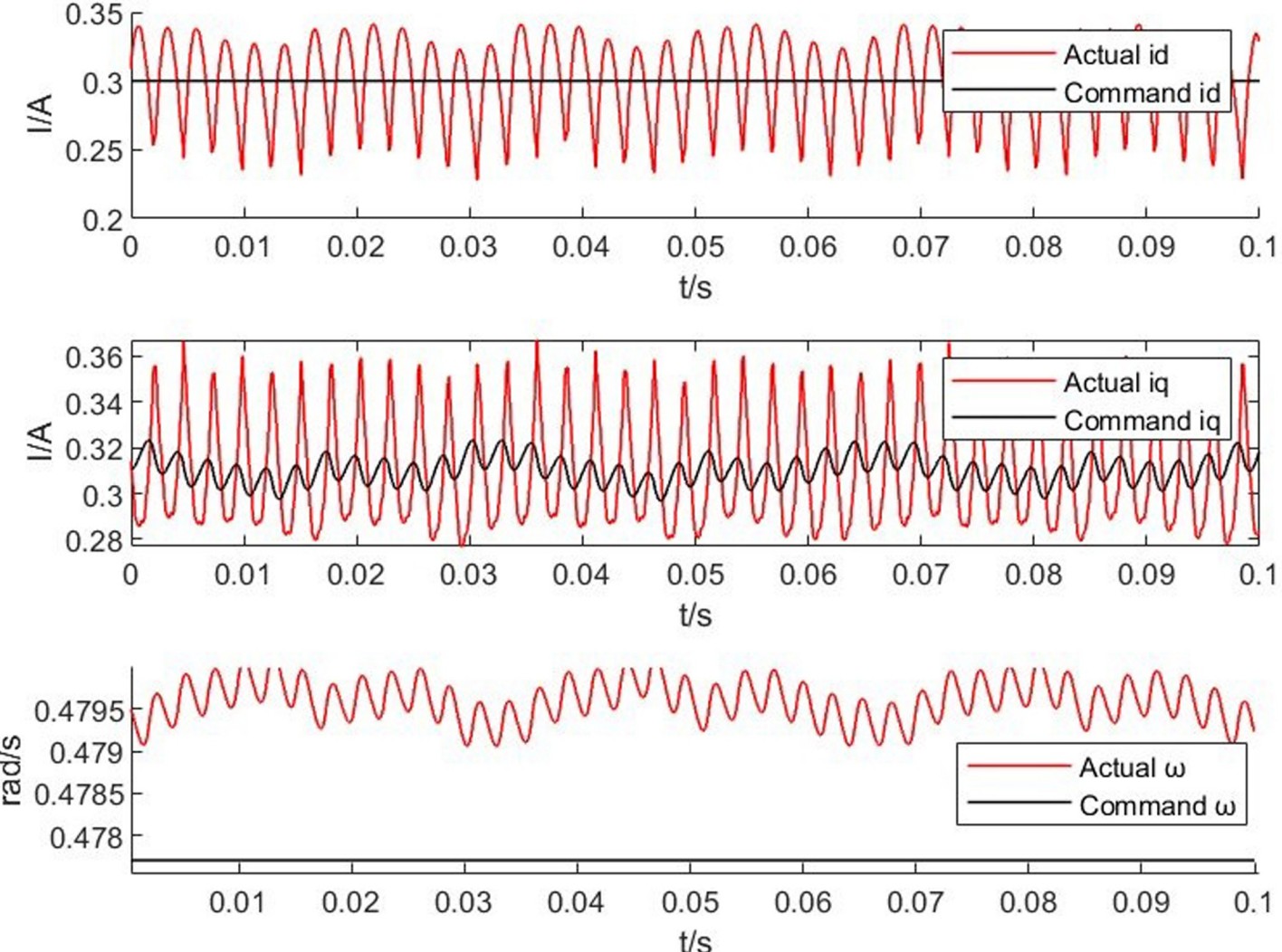

**Fig 15. Motor simulator 10 *N·m* hardware-in-the-loop simulation waveform.**

section. To determine if the permanent magnet synchronous motor simulator can faithfully replicate the port characteristics of the genuine motor module under the three sudden load situations involving the output waveforms of the motor simulator are still compared to the output waveforms of the motor module.

Figs 18–20 show that the straight shaft current is stable at a value of roughly 0.3 A prior to the addition of a sudden load and that the oscillation's amplitude is 0.08 A. The motor simulator's straight shaft current tracking command value is accurate and stable, and it meets with all control criteria. After a clear fluctuation when a sudden loading condition occurs, the direct shaft current is stabilized at around 0.3 A once more. The adjustment times are all less than 0.02 s, and the relative errors are 1.238%, 0.793%, and 0.506%, respectively. These errors are all less than 5%, satisfying the requirements of the motor control strategy test.

From Figs 18–20, it can be found that the cross-axis current $i_q$ is stable at 0 A, 0.32 A and 0.61 A respectively before the sudden load addition, and the oscillation amplitude is ±0.08 A. The stability and accuracy of the cross-axis current tracking command value of the motor simulator meet the control requirements. When the three sudden load conditions occur, the

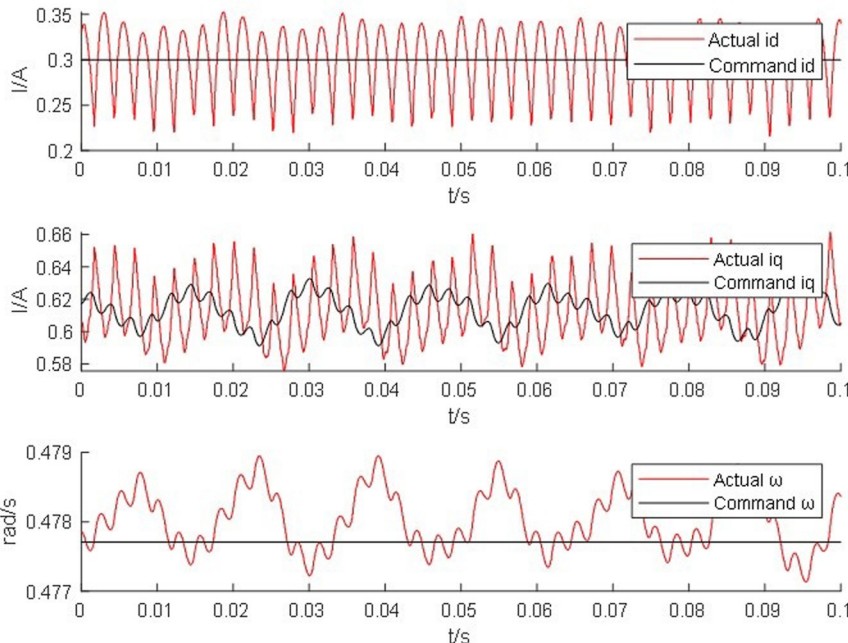

**Fig 16. Waveform of 20 $N\cdot m$ hardware-in-the-loop simulation of the motor simulator.**

waveform of the cross-axis current $i_q$ of the motor simulator and the actual motor module immediately rises and then reverts to a steady-state. The $i_q$ overshoot of the actual motor module is higher than the cross-axis current of the motor simulator, and the regulation time of the motor simulator is less than that of the actual motor module. When the load is suddenly added, the regulation time of the cross-axis current in all three sudden load conditions is less

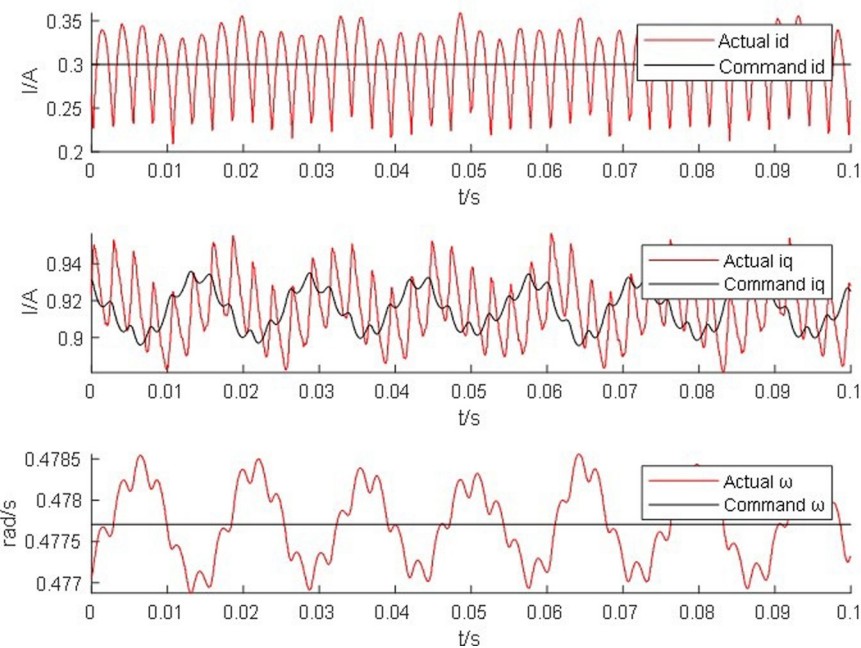

**Fig 17. Motor simulator 30 $N\cdot m$ hardware-in-the-loop simulation waveform.**

**Table 2. The steady state simulation results.**

| Steady conditions | 0 N·m | 10 N·m | 20 N·m | 30 N·m |
|---|---|---|---|---|
| id | 0.299947669 | 0.300245628 | 0.300537588 | 0.299775353 |
| idref | 0.3 | 0.3 | 0.3 | 0.3 |
| iq | 0.0070314 | 0.31002163 | 0.61338548 | 0.91616498 |
| iqref | 0.00670023 | 0.3100582 | 0.61345983 | 0.91550320 |
| w | 0.47772384 | 0.4795983 | 0.47802748 | 0.477729673 |
| wref | 0.477707 | 0.477707 | 0.477707 | 0.477707 |

than 0.05 s, and the relative errors are -2.665%, -0.005%, and -0.013%, respectively, which errors are all less than 5%, and the adjustment times are all less than 0.03 s, meeting the requirements of the motor control strategy test.

From Figs 18–20, it can be found that the angular velocity of the motor simulator is stable at 0.477 rad/s, 0.481 rad/s, and 0.480 rad/s under the three sudden load conditions, and the angular velocity of the motor simulator is stable at 0.465 rad/s, 0.469 rad/s, and 0.468 rad/s after the sudden load, and the difference of the angular velocity amplitude before and after the sudden load is only 0.465 rad/s, 0.469 rad/s, and 0.468 rad/s. The difference between the angular velocity amplitude before and after the sudden load is only 0.012 rad/s. After the numerical conversion, the speed of the motor simulator after the sudden load is 1949.399 rpm, 1966.168 rpm, and 1961.976 rpm, respectively. The average absolute errors of the motor simulator speed and the set speed are -2.530%, -1.691%, and -1.9012% respectively, which errors are less than 5%, and the adjustment time is less than 0.03 s, meeting the requirements of the motor control strategy test. The Transient State Simulation results are shown in Table 3.

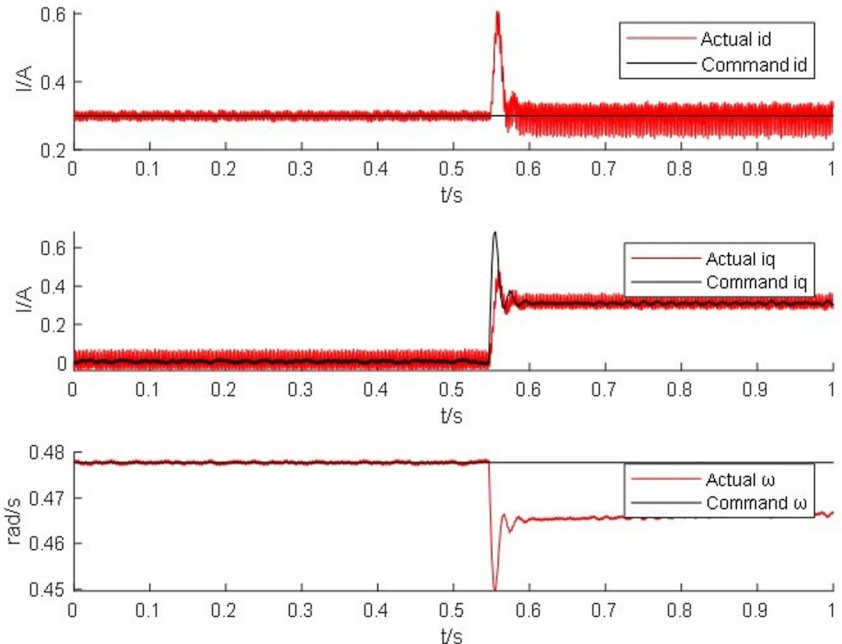

**Fig 18. Real-time simulation waveform of the motor simulator for sudden load condition 1.**

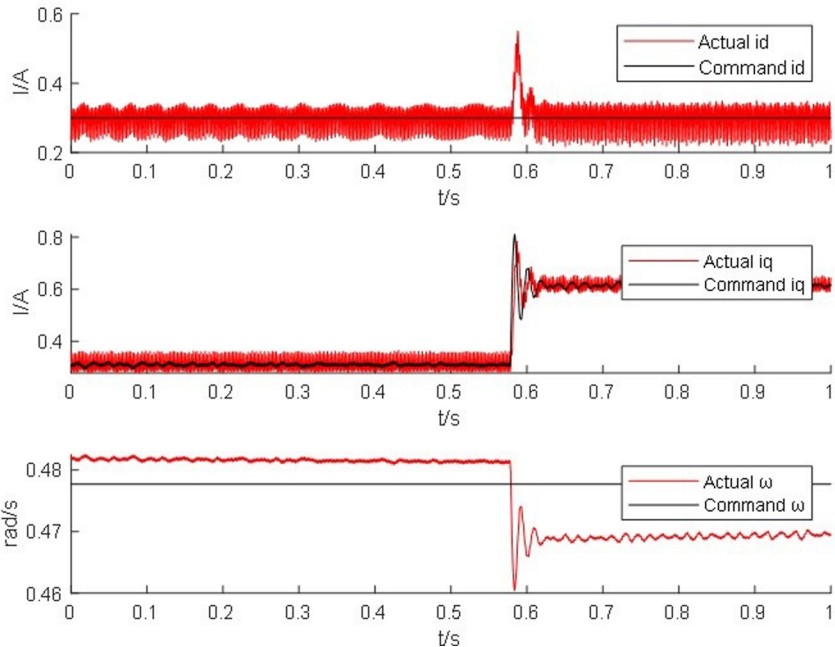

**Fig 19. Real-time simulation waveform of the motor simulator for sudden load condition 2.**

## 6 Conclusion

In this research, a general-purpose motor simulator for motor controller and motor control plan testing is introduced. The following are the contributions of this paper:

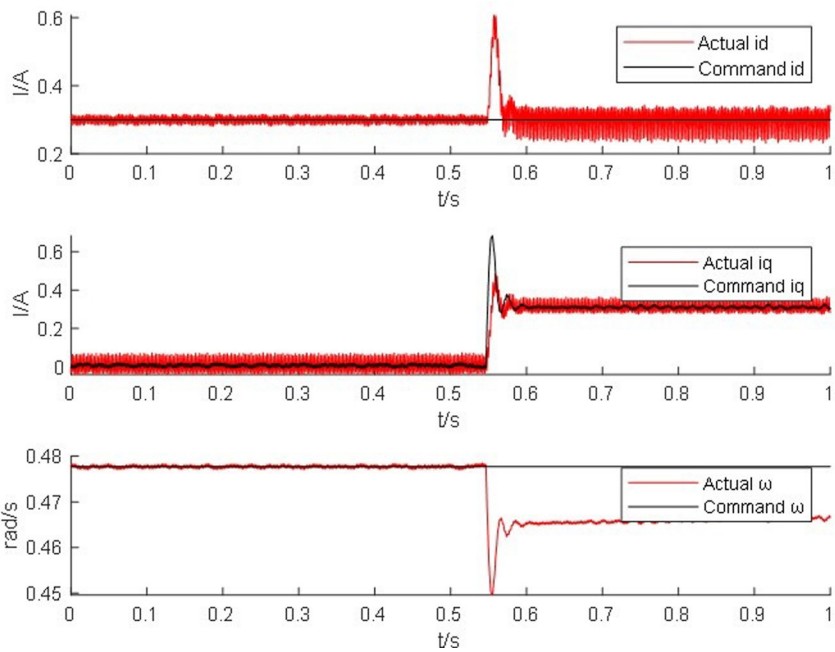

**Fig 20. Real-time simulation waveform of the motor simulator for sudden load condition 3.**

**Table 3. The transient state simulation results.**

| Transient conditions | 0–10 N·m | 10–20 N·m | 20–30 N·m |
|---|---|---|---|
| id | 0.303712996 | 0.302378947 | 0.301517957 |
| idref | 0.3 | 0.3 | 0.3 |
| iq | 0.143839582 | 0.437296177 | 0.745476677 |
| iqref | 0.147777681 | 0.437950046 | 0.745572618 |
| w | 0.477690686 | 0.476312699 | 0.475301769 |
| wref | 0.477707 | 0.477707 | 0.477707 |

Inverse magnetization motor look-up table model using the Flux Linkages as the state quantity is in this article. By using a finite element simulation, the data of the motor's nonlinear features, such as magnetic saturation and spatial harmonics, are acquired and offline saved in the motor model as a look-up table model for use. The permanent magnet synchronous motor's inverse magnetization look-up table model combines the benefits of a simple mathematical model with high fidelity and detail in illustrating the motor's nonlinear characteristics.

## Supporting information

**S1 Table. The data for Flux linkage-current three-dimensional look-up table model.**
(XLSB)

**S2 Table. The data for motor simulator 0 N·m hardware-in-the-loop simulation waveform.**
(XLSB)

**S3 Table. The data for motor simulator 10 N·m hardware-in-the-loop simulation waveform.**
(XLSB)

**S4 Table. The data for motor simulator 20 N·m hardware-in-the-loop simulation waveform.**
(XLSB)

**S5 Table. The data for motor simulator 30 N·m hardware-in-the-loop simulation waveform.**
(XLSB)

**S6 Table. The data for real-time simulation waveform of the motor simulator for sudden load condition 1.**
(XLSB)

**S7 Table. The data for real-time simulation waveform of the motor simulator for sudden load condition 2.**
(XLSB)

**S8 Table. The data for real-time simulation waveform of the motor simulator for sudden load condition 3.**
(XLSB)

## Author Contributions

**Conceptualization:** Yifeng Guo, Limin Huang, Bin Zhong.

**Formal analysis:** Yifeng Guo, Limin Huang, Min Zhang.

**Funding acquisition:** Limin Huang, Bin Zhong.

**Project administration:** Min Zhang, Min He.

**Supervision:** Yifeng Guo, Limin Huang.

**Validation:** Min He, Dakun Fan.

**Writing – original draft:** Min Zhang, Min He.

**Writing – review & editing:** Yifeng Guo, Limin Huang, Bin Zhong.

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
