## [Decision Letter · Decision Letter 0]

7 Nov 2023

PONE-D-23-24308A Versatilely High Fidelity Electric Machines Emulator for Rapid Testing of Motor ControllerPLOS ONE

Dear Dr. Huang,

Thank you for submitting your manuscript to PLOS ONE. After careful consideration, we feel that it has merit but does not fully meet PLOS ONE’s publication criteria as it currently stands. Therefore, we invite you to submit a revised version of the manuscript that addresses the points raised during the review process.

ACADEMIC EDITOR: The reviewers recommend reconsideration the manuscript with revision and modification. I invite the authors to resubmit the manuscript after addressing the comments raised by the reviewers.

We look forward to receiving your revised manuscript.

Kind regards,

Dhanamjayulu C, Ph.D & Post.Doc

Academic Editor

PLOS ONE

“Limin Huang received Funded:Sichuan Province Science and Technology Support Program (2023YFQ0092), which provided support in terms of experimental setup. Chengdu Science and Technology Program (2022-YF05-01393-SN), which provided support in data collection and theoretical analysis. Sichuan College Students Innovation and Entrepreneurship Training Program (S202211079055), which Provided support in cultivating students' innovative abilities.”

“We would like to give great appreciation to the support of Sichuan Province Science and Technology Support Program (2023YFQ0092), Chengdu Science and Technology Program (2022-YF05-01393-SN), and Sichuan College Students Innovation and Entrepreneurship Training Program (S202211079055).”

“Limin Huang received Funded:Sichuan Province Science and Technology Support Program (2023YFQ0092), which provided support in terms of experimental setup. Chengdu Science and Technology Program (2022-YF05-01393-SN), which provided support in data collection and theoretical analysis. Sichuan College Students Innovation and Entrepreneurship Training Program (S202211079055), which Provided support in cultivating students' innovative abilities.”

6. We note that Figure 13 and 14 in your submission contain copyrighted images. All PLOS content is published under the Creative Commons Attribution License (CC BY 4.0), which means that the manuscript, images, and Supporting Information files will be freely available online, and any third party is permitted to access, download, copy, distribute, and use these materials in any way, even commercially, with proper attribution. For more information, see our copyright guidelines: http://journals.plos.org/plosone/s/licenses-and-copyright.

1. You may seek permission from the original copyright holder of Figures 13 and 14 to publish the content specifically under the CC BY 4.0 license.

Additional Editor Comments:

The reviewers recommend reconsideration the manuscript with revision and modification. I invite the authors to resubmit the manuscript after addressing the comments raised by the reviewers.

Reviewers' comments:

Reviewer's Responses to Questions

**Comments to the Author**

1. Is the manuscript technically sound, and do the data support the conclusions?

Reviewer #1: No

Reviewer #2: Yes

2. Has the statistical analysis been performed appropriately and rigorously? 

Reviewer #1: No

Reviewer #2: Yes

3. Have the authors made all data underlying the findings in their manuscript fully available?

Reviewer #1: Yes

Reviewer #2: Yes

4. Is the manuscript presented in an intelligible fashion and written in standard English?

Reviewer #1: No

Reviewer #2: Yes

5. Review Comments to the Author

Reviewer #1: Authors have simulated the electrical machine in real-time using Typhoon HIL in which following observations were noted:

1. It is hard to recognize any novelty in presented work. This is because; operation of an electrical machine with their non-linear characteristics has been already reported in many available literatures.

2. Furthermore, Real-time simulation using hardware emulator is also reported [1, 8, 11, 17, 18, 20].

3. HIL results in the manuscripts seem to be obtained from PC simulation using Typhoon software. It should be captured using suitable indicating instrument/DSO.

4. Authors are required to mention the suitable reason for the consideration of Interior Permanent Magnet (IPM) machine in their work.

5. Authors are also required to add more recent and relevant literature in reference section.

Reviewer #2: This paper is focused on a general-purpose motor simulator for motor controller and motor control plan testing. The real-time simulation tool typhoon HIL is used in the study to develop a hardware-in-the-loop simulation platform for an IPM electric machines emulator.

I have the following comments for the authors:

1. In the Steady State Simulation Analysis section I suggest summarizing the results in a table.

2. In the Electric Machines Emulator Transient Simulation Analysis section I suggest summarizing the results in a table.

3. Improving the quality of the figures in the article.

6. PLOS authors have the option to publish the peer review history of their article (what does this mean?). If published, this will include your full peer review and any attached files.

Reviewer #1: No

Reviewer #2: No

---

## [Author Response · Author response to Decision Letter 0]

17 Jan 2024

As kindly suggested by you and the reviews, the paper has been further carefully revised in accordance to the constructive and helpful comments from you and all the reviewers for improving the quality of paper further. The detailed information and responses to reviewers are provided in the “Detailed Response to Reviewers”. 

We would like to give great appreciation to the support for this paper of the Sichuan Provincial Regional Innovation Cooperation Project (2023YFQ0092), the Chengdu City Technology Innovation R&D Project (2022-YF05-01393-SN), and the Sichuan College Students' Innovation and Entrepreneurship Program (S202211079055).

The funders had no role in study design, data collection and analysis, decision to publish, or preparation of the manuscript. They have no known competing financial interests or personal relationships that could have appeared to influence the work reported in this paper. 

We have modified Figure 13, which may have copyright disputes, and Figure 14 has been deleted.

Response to Reviewer #1:

Comment 1 and Comment 2:

1. It is hard to recognize any novelty in presented work. This is because; operation of an electrical machine with their non-linear characteristics has been already reported in many available literatures.

2. Furthermore, Real-time simulation using hardware emulator is also reported [1, 8, 11, 17, 18, 20].

Response 1 and 2:

The objective of this study is to develop a simulator that can simulate the behavior of a motor in real time by combining the proposed motor model with typhoon. The simulator serves as a control object for the motor controller and helps the developer to develop the related control program in advance. Although many methods for describing motor nonlinearities have been reported in existing studies, the magnetic chain-based motor modeling method proposed in this paper characterizes the real operating characteristics of IPM motors, improves the simulation accuracy and efficiency of the model motor model, and the method has low computational complexity and high real-time performance. The magnetic chain-based motor model can be deployed on in the semi-physical simulation equipment with the advantages of stability and high reliability.

Comment 3:

3. HIL results in the manuscripts seem to be obtained from PC simulation using Typhoon software. It should be captured using suitable indicating instrument/DSO.

Response 3:

The result graph is drawn from the data exported by the oscilloscope through matlab, and no screenshots of the oscilloscope are retained during the experimental phase. The resulting graph proves that the simulator accurately represents the real behavior of the motor, and we will incorporate your suggestions in subsequent studies.

Comment 4:

Authors are required to mention the suitable reason for the consideration of Interior Permanent Magnet (IPM) machine in their work.、

Response 4:

The detailed parameters of the IPM motor in this study come from Toyota's open source motor data, and IPM is widely used in electric vehicles, so this is the motivation for choosing the IPM motor in this article. We will redescribe this part in Chapter 3 of the paper.

Comment 5:

Authors are also required to add more recent and relevant literature in reference section.

Response 5:

We have added some recent references to the paper.

Response to Reviewer #2:

Comment 1:

1. In the Steady State Simulation Analysis section I suggest summarizing the results in a table.

Response 1

We summarize the results of the steady-state simulation in Table 2.

Comment 2:

2. In the Electric Machines Emulator Transient Simulation Analysis section I suggest summarizing the results in a table.

Response 2

We summarize the results of the transient -state simulation in Table 3.

Comment 3:

3. Improving the quality of the figures in the article.

Response 3

We have redrawn the experimental results to improve picture clarity.

As kindly suggested by you and the reviews, the paper has been further carefully revised in accordance to the constructive and helpful comments from you and all the reviewers for improving the quality of paper further. The detailed information and responses to reviewers are provided in the “Detailed Response to Reviewers”. 

We would like to give great appreciation to the support for this paper of the Sichuan Provincial Regional Innovation Cooperation Project (2023YFQ0092), the Chengdu City Technology Innovation R&D Project (2022-YF05-01393-SN), and the Sichuan College Students' Innovation and Entrepreneurship Program (S202211079055).

The funders had no role in study design, data collection and analysis, decision to publish, or preparation of the manuscript. They have no known competing financial interests or personal relationships that could have appeared to influence the work reported in this paper. 

We have modified Figure 13, which may have copyright disputes, and Figure 14 has been deleted.

---

## [Editor Report · Decision Letter 1]

9 Feb 2024

A Versatilely High Fidelity Electric Machines Emulator for Rapid Testing of Motor Controller

PONE-D-23-24308R1

Dear Dr,

We’re pleased to inform you that your manuscript has been judged scientifically suitable for publication and will be formally accepted for publication once it meets all outstanding technical requirements.

Kind regards,

Dhanamjayulu C, Ph.D & Post.Doc

Academic Editor

PLOS ONE

Additional Editor Comments (optional):

The authors have revised the article properly

The article can be accepted the present form
---

## [Editor Report · Acceptance letter]

27 Feb 2024

PONE-D-23-24308R1 

PLOS ONE

Dear Dr. Huang, 

I'm pleased to inform you that your manuscript has been deemed suitable for publication in PLOS ONE. Congratulations! Your manuscript is now being handed over to our production team.

Kind regards, 

on behalf of

Dr. Dhanamjayulu C 

Academic Editor

PLOS ONE